# Progress in Alternative Strategies to Combat Antimicrobial Resistance: Focus on Antibiotics

**DOI:** 10.3390/antibiotics11020200

**Published:** 2022-02-04

**Authors:** Jayaseelan Murugaiyan, P. Anand Kumar, G. Srinivasa Rao, Katia Iskandar, Stephen Hawser, John P. Hays, Yara Mohsen, Saranya Adukkadukkam, Wireko Andrew Awuah, Ruiz Alvarez Maria Jose, Nanono Sylvia, Esther Patience Nansubuga, Bruno Tilocca, Paola Roncada, Natalia Roson-Calero, Javier Moreno-Morales, Rohul Amin, Ballamoole Krishna Kumar, Abishek Kumar, Abdul-Rahman Toufik, Thaint Nadi Zaw, Oluwatosin O. Akinwotu, Maneesh Paul Satyaseela, Maarten B. M. van Dongen

**Affiliations:** 1Department of Biological Sciences, SRM University-AP, Guntur District, Amaravati 522240, India; saranya_shekharan@srmap.edu.in; 2Department of Veterinary Microbiology, NTR College of Veterinary Science, Sri Venkateswara Veterinary University, Gannavaram 521102, India; p_anandkumar@yahoo.com; 3Department of Veterinary Pharmacology and Toxicology, College of Veterinary Science, Sri Venkateswara Veterinary University, Tirupati 517502, India; raogs64@yahoo.com; 4Department of Mathématiques Informatique et Télécommunications, Université Toulouse III, Paul Sabatier, INSERM, UMR 1295, 31000 Toulouse, France; katia_iskandar@hotmail.com; 5INSPECT-LB: Institut National de Santé Publique, d’Épidémiologie Clinique et de Toxicologie-Liban, Beirut 6573, Lebanon; 6Faculty of Pharmacy, Lebanese University, Beirut 6573, Lebanon; 7IHMA Europe Sàrl, 1870 Monthey, Switzerland; shawser@ihma.com; 8Department of Medical Microbiology, Infectious Diseases, Erasmus University Medical Centre (Erasmus MC), 3015 GD Rotterdam, The Netherlands; j.hays@erasmusmc.nl; 9Department of Epidemiology, High Institute of Public Health, Alexandria University, Alexandria 21544, Egypt; high.Yaramohsen@alexu.edu.eg; 10Infectious Disease Clinical Pharmacist, Antimicrobial Stewardship Department, International Medical Center Hospital, Cairo 11511, Egypt; 11Faculty of Medicine, Sumy State University, 40007 Sumy, Ukraine; andyvans36@yahoo.com (W.A.A.); drakelin24@gmail.com (A.-R.T.); 12Research Coordination and Support Service, National Institute of Health (ISS) Viale Regina -Elena, 299, 00161 Rome, Italy; mariajose.ruizalvarez@iss.it; 13Infectious Diseases Institute (IDI), College of Health Sciences, Makerere University, Kampala 7072, Uganda; nanonosylvie@gmail.com; 14School of Medicine, University of Leeds, Leeds LS2 9JT, UK; epnansmedic01@gmail.com; 15Department of Health Science, University “Magna Graecia” of Catanzaro, 88100 Catanzaro, Italy; tilocca@unicz.it (B.T.); roncada@unicz.it (P.R.); 16ISGlobal, Hospital Clínic-Universitat de Barcelona, 08036 Barcelona, Spain; natalia.rc96.nr@gmail.com (N.R.-C.); morenomorales93@gmail.com (J.M.-M.); 17James P Grant School of Public Health, BRAC University, Dhaka 1212, Bangladesh; rohul.amin@g.bracu.ac.bd; 18Nitte (Deemed to be University), Division of Infectious Diseases, Nitte University Centre for Science Education and Research, Deralakatte, Mangalore 575018, India; krishnakumarb@nitte.edu.in; 19Department of Microbiology, Kasturba Medical College, Manipal, Manipal Academy of Higher Education, Manipal 576104, India; abishek77@hotmail.com; 20Oxford University Hospitals NHS Foundation Trust, Oxford OX3 9DU, UK; dr.thaintnadizaw@gmail.com; 21Department of Microbiology and Biotechnology Centre, Maharaja Sayajirao University of Baroda, Vadodara 390002, India; tosinakinwotu@gmail.com; 22Environmental and Biotechnology Unit, Department of Microbiology, University of Ibadan, 200132 Ibadan, Nigeria; 23Acharya Institute of Technology, Soladevanahalli, Bengaluru 560107, India; maneeshpaul@acharya.ac.in; 24AMR Insights, 1017 EG Amsterdam, The Netherlands; maarten@amr-insights.eu

**Keywords:** antimicrobial resistance, antibiotic resistance, antibiotic alternatives, enzyme inhibitors, antimicrobial peptides, bacteriophages, antimicrobial-resistant enzymes, anti-plasmids, biofilms, anti-virulence

## Abstract

Antibiotic resistance, and, in a broader perspective, antimicrobial resistance (AMR), continues to evolve and spread beyond all boundaries. As a result, infectious diseases have become more challenging or even impossible to treat, leading to an increase in morbidity and mortality. Despite the failure of conventional, traditional antimicrobial therapy, in the past two decades, no novel class of antibiotics has been introduced. Consequently, several novel alternative strategies to combat these (multi-) drug-resistant infectious microorganisms have been identified. The purpose of this review is to gather and consider the strategies that are being applied or proposed as potential alternatives to traditional antibiotics. These strategies include combination therapy, techniques that target the enzymes or proteins responsible for antimicrobial resistance, resistant bacteria, drug delivery systems, physicochemical methods, and unconventional techniques, including the CRISPR-Cas system. These alternative strategies may have the potential to change the treatment of multi-drug-resistant pathogens in human clinical settings.

## 1. Introduction 

Multiple antimicrobials have been developed and marketed over many decades with one common objective–to treat and cure mild to serious infections. A serendipitous discovery of penicillin in the late 1920s led to the discovery of diverse antimicrobials: including multiple advances on the ground-breaking antibiotic penicillin itself. Research has also led to new anti-viral drugs for the treatment of previously impossible to treat diseases, including AIDS, amongst others. Similarly, antifungal (also known as anti-mycotic) and anti-parasitic agents have emerged as crucial tools to combat infection.

While these antimicrobials have played a critical role in improving our health and life expectancy, their utility has largely been compromised by the emergence of the phenomenon of antimicrobial resistance (AMR) in response to antimicrobials. The major consequence of AMR is that, as antimicrobials lose their efficacy, infections become more difficult to treat and significantly increase the risk of disease transmission, severe illness, and death. Notably, AMR comes in all shapes and sizes. Increasingly, many organisms are multi-drug-resistant (MDR) and even more challenging to treat. Of critical concern, though, are organisms that are extensively drug-resistant (XDR) and pan-resistant (PDR) which are practically impossible to treat with standard therapies. The WHO has officially recognized that antibiotics and other antimicrobial medications are becoming increasingly ineffective as a result of AMR, and illnesses have become more difficult or even impossible to treat [1] The OIE (World Organization for Animal Health) international committee unanimously adopted the list of Antimicrobial Agents of Veterinary Importance at its 75th general session in May 2007 [2].

AMR has significant effects in terms of pharmaco-economic burdens. For example, the Infectious Disease Society of America (IDSA) published a study that reported the costly burden of AMR among the U.S Medicare population, which showed that, in 2017, infections caused by bacteria resistant to various antibiotics cost the US $1.9 billion in health care costs, 400,000 days in the hospital, and caused 10,000 deaths among the elderly [3]. This was preceded by the 2014 UK Review on Antimicrobial Resistance chaired by Lord Jim O’Neill that revealed 700,000 annual deaths from resistant infections, expected to rise to 10 million annual deaths at a total cost of $100 trillion in economic output by 2050 if we do not find proactive solutions to prevent the rise in drug resistance [4]. In a first-ever first comprehensive assessment of the global burden of AMR based on the statistical analysis of the available data in 2019 from 204 countries, it was estimated that AMR contributes to 1.27 million deaths among the 4.95 million deaths associated with bacterial AMR. The AMR deaths due to resistance were predicted to be the highest in sub-Saharan Africa and lowest in Australasia. Furthermore, it was predicted that methicillin-resistant *Staphylococcus aureus* (MRSA) was responsible for half a million deaths, while the six pathogens *Escherichia coli*, *Staphylococcus aureus*, *Klebsiella pneumoniae*, *Streptococcus pneumoniae*, *Acinetobacter baumannii*, and *Pseudomonas aeruginosa* were attributed to between 50,000 and 100,000 deaths [5].

AMR has no real boundaries and has silently evolved into a global public health issue that threatens populations from high, medium, and low-risk countries. The environment, food production, poverty, health security, and the achievement of the UN’s Sustainable Development Goals (SDGs) will all be affected, emphasizing the need for a multisectoral One Health strategy for curbing AMR [6]. The increased frequencies of AMR, especially among clinically significant ESKAPEE pathogens (*Enterococcus* species, *Staphylococcus aureus*, *Klebsiella pneumoniae*, *Acinetobacter baumannii*, *Pseudomonas aeruginosa*, *Enterobacter* species, and *Escherichia coli*), has put tremendous pressure on the healthcare, veterinary, and agriculture industries, making it one of the world’s most urgent public health concerns [7,8]. Further, in a ‘One Health’ context, the consequences of the spread of AMR bacteria from food animals may have a profound impact on both animal health and public health [9]. Considering the global health impact of AMR bacteria and the need for new antibiotics, new strategies are being implemented to protect and treat MDR, XDR, and PDR infections as even so-called ‘antibiotics of last resort’ are becoming ineffective in clinical settings [10].

This review is to comprehensively highlight various alternative strategies (Figure 1) in the following categories: (1)targeting antimicrobial-resistant enzymes;(2)targeting antimicrobial-resistant bacteria;(3)drug delivery systems;(4)physiochemical methods; and(5)unconventional strategies.

## 2. Historical Perspectives

Antimicrobial medications have revolutionized not just the treatment of infectious diseases but also the human life span. Many of these developments are highlighted in Figure 2. Salvarsan, a syphilis treatment developed by Ehrlich in 1910 [11], was among the world’s first early antimicrobial agents. However, perhaps the best-known antibiotic is penicillin, discovered by Alexander Fleming in 1928, followed by Domagk and other researchers, who synthesized sulfonamides in 1935. Nevertheless, the sulfonamides had some notable safety and efficacy limitations. During the next two decades, a variety of new classes of antimicrobial agents were developed, leading to the so-called ‘golden age of antimicrobial chemo-therapy’. Some examples include streptomycin, an aminoglycoside antibiotic, which was isolated in 1944 from a soil bacterium called *Streptomyces griseus*. Other soil microorganisms yielded chloramphenicol, rifampicin, tetracyclines, macrolides, and glycopeptides (such as vancomycin and, later, teicoplanin). In 1962, the antibacterial agent nalidixic acid, a quinolone antimicrobial, was developed, followed by the cephalosporins that were discovered in the 1960s and rapidly became popular with clinicians. Since then, antimicrobial agents have continued to improve with respect to intrinsic efficacy and spectrum of activity. A very good example of this is the carbapenem class, which was designed to possess broad spectrum antibacterial activity, including against pathogens that exhibited resistance to other classes of antibiotics at that time [12].

Though numerous companies originally competed in the development of newer antimicrobial agents, the number of novel antimicrobials has been steadily declining in recent years, with only a few antimicrobial agents of new classes becoming accessible. At the beginning of the 1980s, several companies lost interest in the development of antimicrobial agents as they did not guarantee continuous market expansion and profits [13]. When coupled with increasingly widespread global AMR, the situation ensured that non-traditional strategies became potentially attractive as new therapeutic avenues. In contrast to past decades, the majority of the companies now involved in antimicrobial agent development are small- to medium-size pharmaceutical companies. The positive news is that multiple alternative strategies, such as anti-virulence strategies, microbiome-modifying strategies, immunomodulators, modified phages, and probiotics, are now being pursued. It should be noted that phage therapy dates back many decades in Europe, with bacteriophages used in Russia for treating soldiers who had dysentery or gangrene in the 1940s [14].

## 3. Conventional Antibiotics to Combination Therapy

### 3.1. Antibiotics Groups and Their Mode of Action

As listed in Table 1, antibiotics are classified into groups based on their class of molecules and targets/primary mode of action; antimicrobial targets include cell membranes, cell walls, protein synthesis, DNA or RNA synthesis, and biological metabolic compound synthesis.

### 3.2. Understanding AMR Mechanisms and the Use of Inhibitors

Increasing global antibiotic resistance in bacteria is driven by a variety of mechanisms, both ancestrally intrinsic to a pathogen’s biology or by emerging mechanisms, triggered by the steadily growing selective pressure exerted by the overuse and misuse of antibiotics in the human, veterinary, and agricultural sectors.

At least four mechanisms of bacterial antimicrobial resistance (Figure 3) have been well defined:(1)Enzymatic degradation of antibiotics, e.g., bacterial synthesis of β-lactamases that degrade the β-lactam class of antibiotics;(2)Modification of the antibiotic target, i.e., the target becomes modified so that the antibiotic is no longer able to bind to its site of action;(3)Control of drug entry through mutations in bacterial cell wall porin molecules and membrane modifications;(4)Activation of efflux pump systems that are able to pump antibiotics out of the cell before antibiotic–target interactions take place.

Understanding the mechanisms of antibiotic resistance has helped in the development and use of several resistance mechanism inhibitors, such as: (a)AMR gene silencers, which silence the AMR genes, e.g.: CRISPR-Cas system(b)Ribosomal inhibitors, which bind with ribosomal subunits and alter the protein production so that the bacteria cannot fight by proteins; and(c)efflux pump inhibitors.

### 3.3. Combination Therapy

Antibiotic combination therapy entails prescribing two or more antibiotics simultaneously, with the goal of obtaining synergistic activity that may be more beneficial for the treatment of patients. The term ‘antibiotic synergy’ is defined as the enhanced effect of one antibiotic with another when combined at the optimal ratio [21]. Different combinations exist (for example, antibiotic + antibiotic or antibiotic + biocide, antibiotic + small molecule and antibiotic + enzyme inhibitor). We need to bear in mind that, for several decades, this combination principle has been tested to find a combination that is translated from in vitro to in vivo and finally into clinical combination. Not many have succeeded, although a few have, especially among the β-lactam with β-lactamase inhibitor combination, and an aminoglycoside combination.

#### 3.3.1. Antibiotic Combinations

Traditionally, there are several antibiotic combinations that have been used to combat MDR infections. In particular, the broad spectrum and synergy of β-lactam antibiotics allows them to synergistically combine with several other groups of antibiotics. For example, the use of a β-lactam antibiotic in combination with an aminoglycoside antibiotic is a well-studied combination, being widely used for the treatment of various Gram-negative bacterial infections [22]. Essentially, the impairment of peptidoglycan synthesis by β-lactam antibiotics potentiates aminoglycosides by rapidly increasing their intracellular concentration in the bacterial cell [23,24]. Other commonly used clinical antibiotic combinations include β-lactam/fluoroquinolone and β-lactam/tetracycline combinations. The aminoglycoside antibiotic amikacin also displayed synergy when combined with colistin, but its therapeutic benefits in treating clinical infections are challenging due to excessive renal toxicity [25]. On the other hand, certain combinations of antibiotics have long been believed to be more effective than using a single antibiotic; however, the real effectiveness of those antibiotics’ combinations is not clear as the resistance mechanisms continue to evolve [22,26]. Therefore, the effectiveness of those antibiotics pairs or any emerging alternative strategy in multi-drug environments should be continuously assessed to combat AMR [27].

#### 3.3.2. Antibiotic Combination with β-Lactamase INHIBITORS

For β-lactam antibiotic-resistant infections, the combination of β-lactamase inhibitors (such as sulbactam, clavulanic acid, and tazobactam) with β-lactam antibiotics helps restore the action of β-lactam antibiotics. This specific type of combination is referred to as a β-lactam/β-lactamase inhibitor (BLBLI) combination. In recent years, new Bis, such as avibactam, relebactam, taniboractam, tazobactam, vaborbactam, enmetazobactam, and zidebactam, have been used to optimize antibiotic therapy for resistance to the newer β-lactam antibiotics (carbapenem resistance), or as carbapenem-sparing antibiotic combinations [28]. For example, the aztreonam–avibactam combination is currently effective against NDM (New Delhi metallo-beta-lactamase), VIM (Verona Integron-encoded metallo-β-lactamase), and IMP-producing bacteria (inactivate imipenem) and is used in clinical settings to treat carbapenem-resistant Enterobacteriaceae (recently renamed as Enterobacterales). In contrast to avibactam, aztreonam is resistant to the action of metallo-β-lactamases (because the prevalence of carbapenem antibiotic-degrading metallo-β-lactamases is high) [29]. Similarly, fourth generation broad-spectrum cephalosporins, such as cefepime’s activity, is restored in combination with enmetazobactam [30].

During the last ten years, resistance in ESKAPE pathogens has increased exponentially worldwide, and PDR has become widespread in clinical settings [31]. Several antibiotic combinations have been studied for treating PDR infections as there are few remaining ‘drugs of last resort’. Studies have reported the improved treatment efficacy of colistin (a drug of last resort) in combination with rifampicin or meropenem or tigecycline [32].

#### 3.3.3. Combination of Antibiotics with Biocides

Combination of antibiotics with biocides (disinfectants, antiseptics, and preservatives), although theoretically effective, has received little interest [33]. In a study to investigate the effect of combining antibiotics and biocides using three antibiotics and seven biocides having different modes of action tested against *P. aeruginosa*, the results demonstrated different combinations of effects varying between synergism and antagonism [34]. Future research should explore the potential evolutionary consequences of the physiological interaction between antibiotics and biocides as these combinations showed broad potential for countering AMR using existing agents.

## 4. Strategies Targeting Antimicrobial-Resistant Enzymes

### 4.1. Enzyme Inhibitors

Enzyme inhibitors are low sub-atomic weight synthetic particles that can diminish or completely inhibit enzyme catalytic activity either irreversibly or reversibly. This is exemplified by the inhibitors of monoamine oxidases (MAO) and the cholinesterases (ChE), which are used for several pharmacological purposes [35]. Enzymes remain ideal targets for therapeutic drugs as modifying the chemical action of an enzyme has a proven positive impact on the course of disease. Indeed, 47% of all the current medications inhibit enzyme targets, even with the increase in the use of medications for receptors to adjust signals from outside the cell [35].

Several antibiotics inhibit enzymes, and correspondingly, many bacterial enzymes play a key role in the development of resistance to these antibiotics. Many antibiotics are developed to target enzymes, and the development of resistance occurs when there are structural changes in those target enzymes or enzymatic modifications in the elements affected by antibiotics. The enzymes that usually serve as targets for antibiotics include enzymes involved in cell wall synthesis, nucleic acid replication, and metabolites. For example, penicillin binding proteins (PBPs) are components of bacterial cell walls that play a major role in the synthesis of peptidoglycan (major constituent of bacterial cell walls). PBPs catalyze transglycosilation and transpeptidase reactions, which leads to elongation and crosslinking of respective peptide chains. PBPs are major targets for most of the antibiotics currently used today. These antibiotics act as competitive inhibitors of PBPs and disrupt the synthesis of the bacterial cell wall [36]. Another example of bacterial targets is the type II topoisomerases (DNA gyrase and topoisomerase IV), enzymes that regulate the supercoiling of DNA during replication and transcription. These enzymes serve as targets for antibiotics that are derivatives of quinolones, which bind covalently to the active sites of these enzymes and inhibit replication and transcription [19,36].

A traditional example of a β-lactamase inhibitor is clavulanic acid, a semi-synthetic molecule commonly administered in combination with β-lactam antibiotics to inactivate β-lactamases involved in the degradation of β-lactam antibiotics. The molecule contains a β-lactam ring and is a ‘suicide’ inhibitor of β-lactamases. More recently, several other molecules with inhibitory activity against enzymes that confer antimicrobial-resistant activity have been evaluated as promising natural or recombinant weapons in the fight against clinically relevant antibiotic-resistant pathogens [15,37].

### 4.2. Medicinal Plants and Phytochemicals

Plants have evolved unique mechanisms to protect themselves from microorganisms via natural phytochemicals (secondary metabolites) found in seeds, roots, leaves, stems, flowers, and fruits [38,39,40,41,42]. Further, plants synthesize many structurally different chemicals that possess a specific role in their response to microbial attack [41,43,44]. Therefore, the potential efficacy of plant-derived compounds as drug candidates has attracted the attention of the pharmaceutical and scientific communities, who have evaluated many diverse plant extracts and oils as potential antibacterial and antibiotic resistance-modifying agents [38,39,40]. The screening programs implemented for such novel drug discovery include random, computational, and ethnopharmacological approaches [42].

The medically important plant-derived substances (PDSs) that exert the strongest antimicrobial activity include alkaloids, organosulfur, phenolic compounds, coumarin, and terpenes [43,44]. The in vitro antibacterial activity of PDSs has been shown against a broad range of bacteria, including MDR [41,45,46,47,48,49,50,51,52,53,54,55,56], with the mechanisms of action including: (i) inhibition of bacterial cell wall synthesis, (ii) inhibition of bacterial physiology, (iii) modulation of antibiotic susceptibility, (iv) biofilm inhibition, (v) attenuation of bacterial virulence, and (vi) inhibition of efflux pumps [44]. For example, alkaloids and phenolic compounds have an inhibitory effect on the efflux pumps of *E. coli* [57,58], *Staphylococcus aureus* (*S. aureus*) [59,60,61,62,63,64,65,66], and methicillin-resistant *S. aureus* (MRSA) [67,68,69]. Alkaloids also inhibit cell division, protein synthesis, and DNA in *E. coli* [70] and the inhibition of ATP synthase in *Listeria*, *Bacillus*, and *Staphylococcus* spp. [71]. Phenolic compounds inhibit β-ketoacyl acyl carrier protein synthase (*KAS*) III, a key catalyst in bacterial fatty acid biosynthesis, with MIC values of *Enterococcus faecalis* in the range of 128 to 512 μg/mL [72]. The antibacterial mode of action of organosulfur compounds includes the inhibition of sulfhydryl-dependent enzymes, ATP synthase, DNA, and protein synthesis and the destruction of the bacterial membrane of *Escherichia coli*, *Staphylococcus epidermidis*, *Pseudomonas aeruginosa*, *Streptococcus agalactiae*, and *Campylobacter jejuni* [62,73,74]. Coumarin inhibits DNA gyrase of *Staphylococcus aureus* [75,76,77], Methicillin-resistant *Staphylococcus aureus* (MRSA) [78], *E. coli* [77], *P. aeruginosa* [77], and *Helicobacter pylori* [75], and terpenes lead to cell membrane disturbance of *S. aureus*, *P. aeruginosa*, *E. coli*, and *H. pylori* [79,80]. Additionally, the efficiency and efficacy of the antibiotic ceftiofur against important mastitis-causing bacteria in bovines was reported to be enhanced in combination with phytochemical phosphorylcholine [81]. PDSs have demonstrated a potent antimicrobial activity, either alone or when combined with antibiotics, and have promising potential in the development of novel drugs to fight AMR [44]. Finally, the use of phytochemical products with proven antimicrobial properties may be a viable alternative to the use of antibiotic-based growth promoters as feed additives in livestock and poultry farming [82].

### 4.3. Small Molecules-Improved Chemical Entities (ICE)

Natural products and their semi-synthetic derivatives, for example, β-lactam antibiotics, aminoglycosides, tetracyclines, and macrolides, are the mainstays of the current antibiotic therapies. However, the effectiveness of these antibiotics is now threatened by the global spread of multi-drug-resistant pathogens [83,84,85]. Fortunately, advances in genomics and innovative technologies offer the possibility of re-examining discarded chemical scaffolds, revitalizing natural product programs, and ultimately finding new leads [84,85]. Newer direct-acting small molecules may be generated as improved derivatives of older antibiotics or new chemicals with new targets and novel mechanisms of action [83,86]. The newer small molecules include three groups comprising (i) synthetic and natural antimicrobial peptides (AMPs), (ii) natural chemicals, and (iii) inhibitors, such as LpxC and LpxA [83,87].

Antimicrobial peptides (AMPs) are a naturally abundant and diverse group of antibacterial agents [88,89,90]. These direct-acting small molecules have broad-spectrum activity, a rapid and sustained bactericidal effect, and are highly selective [90]. Their limitations are related to their short plasma half-life due to proteolytic degradation and issues with toxicity [90,91]. Antimicrobial peptidomimetic compounds, for example, α-peptoids, mimic the structure and biological activity of natural AMPs [90] yet offer the advantage of overcoming some of the functional issues associated with natural AMPs, such as stability in the presence of biological matrices [91,92].

Inhibitors targeting essential enzymes may be of use; for example, LpxC is an enzyme of lipid A biosynthesis in Gram-negative bacteria and a promising target for developing antibiotics that selectively target Gram-negative pathogens [92]. In the mid-1990s, a clinical candidate exhibiting LpxC inhibitory activity and low MIC values against a wide range of Gram-negative bacteria failed in phase 1 human clinical trial due to the toxicity of the product [83,85]. A growing body of knowledge and experience may help overcome some of the current hurdles, such as undesired effects caused by the common structural elements of enzyme inhibitors [85].

Most of the direct-acting new molecules now studied target Gram-negative bacteria, such as Enterobacterales and non-fermenters, and few broadly target both Gram-negative and Gram-positive bacteria [83]. More push and pull incentives are needed to revitalize the clinical pipeline and find novel therapies [83,85].

Besides, recent studies highlight the role of small molecules from mixed microbial communities in hampering the effects of the antibiotics and/or rescuing antibiotic molecules from degradation [93,94,95,96,97,98].

### 4.4. Essential Oils

‘’Essential oils” (EOs) are mixtures of volatile chemical compounds synthesized from different plant parts during secondary metabolism [99,100,101]. The term ‘essential oil’ was first used by Paracelsus von Hohenheim, a medieval Swiss physician [102]. EOs include terpenes, aldehydes, phenolic, terpenoids, and other aromatic constituents that have demonstrated antimicrobial activities [99,100,101,103,104,105,106,107,108,109,110,111,112,113,114]. EOs mainly contribute to the disruption of the bacterial cell membrane and inhibition of the efflux pump responsible for certain AMR in Gram-negative bacteria [100,105,114,115,116,117]. Other documented modes of action include inhibition of the peptidoglycan layer synthesis of bacterial cell walls by binding to PBPs for Gram-positive bacteria [105,114,115,116,117]. Recent advances in genomics and proteomics demonstrated the ability of EOs to inhibit biofilm formation and quorum sensing (QS) production and increase the expression of oxidative stress proteins [100,105]. Eos, either alone, in combination, or associated with other antibiotics, demonstrated effective antibacterial activity against different pathogens, including MDR bacteria [100,105,118]. More in-depth studies are necessary to discover and identify novel EO compounds that may one day be used in clinical practice [100,103]. EOs that have been extensively studied include cinnamon bark [119,120,121,122,123], lavender [107,124,125], peppermint [126,127], and tea tree oil [128,129,130,131]. Other studied EOs include, but are not limited to, eucalyptus, black pepper, lemongrass, and palmarosa [100,104,132]. Furthermore, the use of antibiotics as growth promoters in livestock/aquaculture production seems necessary in today’s world. Under these circumstances, EOs are ‘green’ and promising as alternatives to the current antibiotic growth promoters used by livestock/aquaculture farmers [133,134]. It has also been documented that EOs exhibit food preservation properties, and various EOs have been tried as food preservatives to prolong the shelf life of meat, meat products, dairy products, vegetables, and fruits [135]. Coupling EOs with nanoparticle technology could potentially facilitate a promising improvement in the chemical stability and solubility of EOs [136]. Nanotechnology potentially allows the delivery of nano-encapsulated EOs to the target site, thereby minimizing toxicity while maximizing EO efficiency. An appreciation of the interplay between the components of crude EOs, the discovery of novel compounds, and the clinical approval of EOs as antimicrobial agents is increasingly gaining importance [100,103].

It is worthwhile to highlight the demonstration of the bacteria developing resistance and tolerance towards EOs; however, cross-resistance to antibiotics was not reported [137]. Treatment of *P. aeruginosa* infection with a sub-inhibitory concentration of cinnamon bark oil or cinnamaldehyde as an adjunctive therapy may potentially induce expression of efflux pumps, and this needs further investigation to ascertain the use of EOs with any antagonistic effects [138].

### 4.5. RNA Silencing

A strategy that also has the potential to generate novel antimicrobials is RNA silencing. RNA silencing is naturally found in bacteria, was first described in 1985, and is now known to be associated with the regulation of many genes. The mechanism involves cis and trans sequences that are complementary to regulatory regions on a single m-RNA (antisense sequence) and which, upon binding, can reversibly block translation. Cis antisense sequences can be found near regulatory regions on a single RNA or may be transcribed from the complementary strand at the same genetic locus. Trans sequences are transcribed from a distant genetic locus and form most of the natural antisense sequence. Synthetic antisense sequences can potentially be developed to repress the translation of enzymes that enable bacteria to resist antibiotics. RNA silencing is applied in the discovery of new antimicrobial compounds, determination of the stringency requirement for those targets, development of highly sensitized antimicrobial screens, and mode of action [139].

RNA silencing may also be applied in the development of antibacterial screening. This enables genes of target interest to be knocked down. For example, 250,000 natural products for *FabF*/*FabH* inhibitors (which prevent bacterial fatty acid biosynthesis pathway) were screened by Merck Research Laboratories using an *S. aureus* strain expressing antisense RNA to *fabF*. RNA silencing can also be applied in detecting the mode of action of novel antibiotics; for example, RNA silencing was used in the discovery of antimicrobial agents involving novel enoyl–acyl carrier protein reductase (*Fabl*) inhibitors [139].

### 4.6. CRISPR-Cas System

CRISPR-Cas (clustered regularly interspersed short palindromic repeats-CRISPR associated protein) is a bacterial adaptive immune system that uses DNA-encoded, RNA-mediated, or DNA-targeting processes to counter the invasion of bacteria by foreign genetic material and mobile genetic elements, such as plasmids and phages [140,141,142]. CRISPR-Cas are genomic engineering tools offering promising new leads as programmable sequence-specific antimicrobials [140,143]. These gene-editing tools can target quantitatively, specifically, and selectively bacterial genomes to reduce or eliminate antibiotic resistance and create new opportunities to treat MDR infections [140,141,142,143,144]. CRISPR-Cas systems can discriminate between pathogenic and commensal bacteria and are potentially capable of selectively removing AMR genes from bacterial populations and bacterial virulence factors, or sensitizing bacteria to an antibiotic by eliminating plasmids harboring antibiotic resistance genes [143,145].

Within all Cas proteins, the ones used that show the most promise for AMR include (i) CRISPR-Cas9, an RNA-guided-DNA cleavage, (ii) dCas9, (iii) nSpCas9:rAPOBEC1, and (iv) Cas13a [146]. CRISPR-Cas offers new potential with respect to AMR [140,141,142,143,144,145,146,147,148,149,150,151]. Further studies are needed to address the limitations while focusing on in vivo experiments [145], such as (1) the delivery issues addressed by the use of phage-delivery and phagemids, conjugative plasmids, to polymeric nanoparticles; (2) the side effects of potential off-target modifications in the host’s genome [145,152,153,154,155].

## 5. Strategies Targeting Antimicrobial-Resistant Bacteria

### 5.1. Lantibiotics and Bacteriocins

The term “lantibiotic” designates gene-encoded peptides that contain unusual amino acids, including the thioether amino acids lanthionine (Lan) and/or methyllanthionine (MeLan), which are formed by post-translational modification and consequently introduce intramolecular cyclic structures in the peptide needed for the specific exporters and for posttranslational modification [156]. Many lantibiotics have been discovered over recent years, and descriptions of different lantibiotics have been published. Notably, all of these substances are produced specifically by Gram-positive bacteria and exert their inhibitory action mainly against this group [156,157]. For the purposes of this publication, it is worth mentioning that the majority of lantibiotics possess some kind of antimicrobial activity, and the designation “lantibiotic” was derived from the term “lanthionine containing antibiotic”.

Lantibiotics can be grouped into type-A and type-B peptides. In general, type-A lantibiotics are elongated, cationic peptides consisting of up to 34 residues in length that show similarities in the arrangement of their Lan bridges [158]. These peptides primarily act by disrupting the membrane integrity of target organisms and include nisin, subtilin, and epidermin. Type-B peptides are globular, up to 19 residues in length, and act through disruption of enzyme function, e.g., inhibition of cell wall biosynthesis. The duramycins produced by *Streptomyces* species, mersacidin and actagardine, are examples of type-B peptides. However, a number of lantibiotics do not fall into either category, suggesting that lantibiotic classification will undoubtedly become more complex as more compounds are discovered [158].

Bacteriocins, a type-A lantibiotic, are proteinaceous or peptidic toxins produced by bacteria. These molecules are able to kill or inhibit closely related bacterial strains or unrelated bacteria but will not harm the original bacteria through specific immunity proteins. Importantly, bacteriocins have a large diversity of structure and function, and are notably stable to heat [159]. Bacteriocins emerged through extensive pharmaceutical drug discovery efforts and have become an important addition to the current armamentarium of agents that can be used in the future to treat serious bacterial infections [160]. Bacteriocins attracted significant interest due to their wide-ranging properties as antimicrobial specialist agents against a variety of organisms, including various bacterial, parasitic, and viral species, and notably also against more complex systems, such as bacterial biofilms [161]. These ribosomally produced peptides are secreted by microbes living in a highly complex polymicrobial climate and are utilized to inhibit neighboring bacterial species, especially closely related species. The variety of distinctive bacteriocins generated by microorganisms means that these toxins possess an expansive range of action. A considerable number of diverse bacteriocins have been identified, and there remains scope for the identification of many more bacteriotoxins. Furthermore, bacteriocins are also useful against microbes that generate anti-toxin mechanisms [162]. The diversity of bacteriotoxins permits a wide scope of biotechnological and drug applications. One of the main areas affected by the use of bacteriocins is the agro-food industry.

### 5.2. Antimicrobial Peptides (AMP)–Including AMP + Antibiotics Combination

Among the different strategies to develop novel and effective antibiotics are AMPs, used either alone or in combination with traditional antibiotics [163,164,165,166,167,168]. AMPs found in nature range between 10 and 50 amino acids, possess an overall cationic charge, and are amphipathic in nature, featuring a similar or even improved antimicrobial activity compared with traditional antibiotics [169,170]. AMPs are ubiquitous and are found in nature in various environments. Typically, AMPs have a role as components of the innate immune system of many terrestrial and/or aquatic organisms. AMPs of bacterial origin are, in natural conditions, part of the bacterial cell protection system, facilitating protection from toxic invasions (e.g., bacteriophages, exogenous molecules). This process affects inflammation and enhances pathogen killing [171]. Moreover, bacterial AMPs warrant “space” to the producing bacteria in the context of complex microbial communities that harbor the same ecological niches [95,172,173,174,175]. AMPs can target a variety of bacteria, both Gram-positive and Gram-negative, through the common mechanisms of traditional antibiotics [94,170,175,176]. Once AMPs have penetrated the bacterial cell wall, they exert further antimicrobial activity by targeting protein biosynthesis, nucleic acids, and/or impairing the cell wall and membrane production [177] 

Extensive research on AMPs has already led to the production of so-called “designer”-like molecules, i.e., synthetic peptides produced by exploitation of knowledge regarding naturally occurring AMPs. In this case, different portions of well-known AMPs are assembled to design tailored AMPs containing unique and desired features [90,170,178,179]. Unfortunately, however, native AMPs used in clinical practice (in human and veterinary fields) suffer from an important drawback associated with their ease of degradation by proteolytic enzymes and the acidic gastric environment. Therefore, topical or subdermal administration is favored, although this is not an efficient delivery of the molecules at the systemic level. To overcome these issues, a variety of approaches to improve AMP stability are being evaluated, such as the production of constrained AMPs, cyclotides, hybrid AMPs, AMP conjugates, AMP mimetics, and immobilized AMPs [180]. Among these, the conjugation of AMPs with traditional antibiotics is generating promising results via the potential synergistic combination of two compounds, enabling effective targeting and killing of several pathogenic-resistant bacteria [181,182]. For example, the conjugation of AMP magainin with vancomycin showed very exciting results against vancomycin-resistant *Enterococci* [183]. Similar results have been obtained by coupling cationic antimicrobial peptide ubiquicidin with chloramphenicol via a glutaraldehyde linker, showing enhanced activity of the antibiotic against *E. coli* and *S. aureus* and a reduced toxicity against human cells [184]. Interestingly, encouraging results have also been obtained by coupling AMPs with antibiotics with the goal of enhancing in situ drug delivery, thereby improving drug specificity and reduced toxicity [185]. This is dependent on the primary mechanism of action of AMPs on bacterial external wall structures. AMPs act more effectively in synergy with antibiotics than when used individually since they are unable to penetrate the microbial cell when the resistance mechanism of the antibiotic relates to membrane modification. Being able to overcome this barrier, AMPs may be able to return activity to antibiotics that were previously rendered ineffective [165].

Nanotechnology-based drug delivery methods (nanocarriers) are a recent innovation that may evolve into a powerful strategy for the efficacious delivery of AMPs. AMP delivery via nanocarriers could be advantageous by protecting peptides against extracellular degradation by proteases and other peptide-hydrolyzing environments. Targeted nanocarriers could also assist in target selectivity and improved drug pharmacokinetic profiles. There are several types of drug delivery systems, such as liposomes, micelles, polymeric nanoparticles, and dendrimers [186,187].

Apart from stability issues and efficacious delivery of AMPs, toxicity is yet another hurdle to this approach. Most AMPs fail preclinical studies due to high in vivo toxicity. Murepavadin (POL7080), a cyclic protegrin analogue, is an example of a cyclic peptide with promising activity against *P. aeruginosa*, which was discontinued in a phase III study for the treatment of *P. aeruginosa* ventilator-associated bacterial pneumonia due to nephrotoxicity and a setback to preclinical studies [165,188,189]. Currently, as of December 2021, another clinical trial is running to evaluate and develop a novel pharmaceutical formulation based on murepavadin for inhalatory use to treat cystic fibrosis patients affected by *Pseudomonas* spp infections. [190]. Despite having promising antimicrobial activity against key human pathogens where the medical need is high, AMPs in clinical trials usually end up being investigated as topical agents due to their instability and toxicity issues, besides the non-favorable pharmacokinetics [191].

### 5.3. Insect Derived Enzymes and AMPs

Many insects produce complex and various families of enzymes both as survival and defense mechanisms. The diversity of insects is enormous, and there is growing evidence that the natural system of insects is particularly dynamic. AMPs produced by insects are smaller in size and contain cationic groups. Fundamentally, four groups of AMPs are found in insects based on their structure and amino acid composition. They include proline-rich peptides (e.g., drosocin, apidaecin, and lebocin), α-helical peptides (e.g., moricin and cecropin), cysteine rich peptides (e.g., defensin and drosomycin), and glycine-rich proteins (e.g., attacin and gloverin) [192]. The major components of innate immunity of insects are cysteine-rich peptides, which are known for their ability to inhibit biofilm formation [193]. Overall, the interaction of peptide and membrane directly promotes antibiofilm/antibacterial activity. It is now well evident that the main immune effector molecules of insects are AMPs. At the same time, due to the conserved biological evolution, bioactive molecules and signaling pathways within the natural system of insects exhibit distinctly more similarity with vertebrates, including humans. Researchers worldwide are in search of novel bioactive molecules with novel mechanisms of action as antimicrobials. Moreover, AMPs may well act synergistically with classical antibiotics for combating various infections. Several factors such as sequence, the charge, the helicity, the amphipathicity, and the overall hydrophobicity of AMPs, are crucial in considering them as effective antimicrobial agents [193,194].

### 5.4. Nanoparticle Based Strategies

Nanoparticles (NPs) are particles whose sizes lie in the range of 1 to 100 nm [195]. NPs are increasingly used as inhibitors of bacterial growth in applications such as coatings for implantable devices/medical materials and the delivery of antibiotics. They may also be used directly as antibacterial agents [196]. Some bulk metals are known to have antibacterial activity against Gram-positive and Gram-negative bacteria; however, there are other metals that are only active in the NP form [197].

The exact mechanism of action through which NPs exert their antimicrobial activity is not yet fully understood, but three processes have been proposed to occur simultaneously. These include oxidative stress, metal ion release, and non-oxidative mechanisms [196,198]. Specifically, these processes result in: (1) the disintegration of the bacterial outer membrane and/or general cell wall damage, (2) the interaction between intra- and extracellular components and ions from the NPs, (3) the production via photocatalysis of reactive oxygen species that damage bacterial structures, (4) the inhibition of DNA synthesis, (5) the inhibition of enzyme activity, and (6) the interruption of energy transduction [199].

The mechanisms through which metallic NPs exert their actions directly depend on their physical characteristics. Depending on their size, NPs can inhibit bacterial growth through bacterial membrane disruption and affect biofilm formation, with small NPs usually showing potent antimicrobial and antibiofilm activity (e.g., Ag, ZnO, Mg, and NO). Shape also influences activity, with rod-shaped particles inhibiting biofilms better than spherical shaped NPs [200]. The particle characteristics and intrinsic factors, such as size, zeta potential, charge, morphology of the surface, and crystal structure, are responsible for the antimicrobial activity of NPs [196].

Several studies have indicated that the relationship between NPs and the external cell wall structures of bacteria are key in the mechanism of action of NPs and that bacterial cell wall composition affects NP activity [201,202,203,204,205]. NPs need to either interact or overcome these structural components, and they can do so through different interactions. Lipopolysaccharide (LPS) in Gram-negative bacteria is a negatively charged outer region that attracts NPs, while, in Gram-positive bacteria, NPs are hypothesized to distribute throughout the teichoic acid net [196].

Due to their physiochemical characteristics, potential antimicrobial applications of NPs include administration by inhalation, oral ingestion, dermal contact, and intravenous injection. One of the main drawbacks regarding using NPs as novel antimicrobials is their toxicity. NPs are generally toxic to eukaryotic cells at concentrations that inhibit bacterial growth, with the long-term effect of NPs in eukaryotic cells and tissues having been assessed through both in vitro and in vivo studies. Interestingly, toxicity could potentially be overcome by targeted delivery of NPs to the infection site, although this strategy influences NPs potential [198,199].

### 5.5. Coinfection Strategies & Probiotic Bacteria against Pathogens

Among the complimentary alternatives to reduce the spread of AMR pathogens in human and veterinary medicine, adoption of coinfection strategies and/or administration of microbial species having probiotic effects is gaining popularity [206]. Probiotics are viable microorganisms whose administration can confer beneficial effects. The strength of probiotics in the fight against AMR relies on preventing the direct selective pressure exerted by antibiotics on pathogenic microorganisms. Probiotics act to prevent infection via a variety of ecological mechanisms, ranging from competition for ecological space, colonization resistance, and nutrients to the production of AMPs with specific bactericidal activity [167]. In this respect, several studies have already demonstrated the capability of probiotics in maintaining a healthy and balanced microbiota composition and improved function [207,208]. This, in turn, contributes to helping maintain ‘optimal’ health, thereby reducing the chance of opportunistic infection. Additionally, probiotic administration is linked with a significant reduction in antibiotic usage and reduction in the selective pressure associated with conventional therapeutic interventions [209].

Probiotics are mostly administered orally in both human and veterinary practice. Microorganisms with probiotic activities include mostly bacteria, although fungi and yeasts are also commonly used. The most common probiotic bacteria are *Lactobacilli* and *Bifidobacteria*. Several studies have demonstrated their ability to adhere to intestinal epithelial cells and survive in acidic environments and fluids, including bile. Furthermore, investigations have demonstrated their ability to kill, or inactivate, a variety of common pathogens, such as *E. coli*, *S. aureus*, *K. pneumoniae*, *S. typhimurium*, *B. subtilis*, and *P. aeruginosa* [210]. Other than *Lactobacilli* and *Bifidobacteria*, the other genera being commonly adopted as probiotics include *Escherichia, Lactococcus, Streptococcus, Enterococcus, and Bacillus*. Although known as the common inhabitant of the human and animal gastrointestinal tract, the genus *Escherichia* also includes species with health-promoting properties. Of these, *E. coli* has been reported to be beneficial for the treatment of constipation, inflammatory bowel disease, colon cancer, and Crohn’s disease [211,212,213]. Additionally, analogous strains of the genus *Lactococcus* (e.g., *Lactococcus lactis* subsp. Lactis) may show important antibacterial activity against pathogenic and spoilage bacteria in milk products, and their ability to adhere to epithelial cells make them good candidates for probiotic interventions [211,214]. Strains of the genera *Streptococcus* and *Enterococcus* are also worthy of note for their probiotic activities, although the opportunistic pathogenic behavior of some of these bacteria prevent their widespread use in human clinical practice, leaving their major application restricted to the veterinary sector [174,215]. Finally, the genus *Bacillus* includes bacteria with demonstrated probiotic features (e.g., *B. subtilis*, *B. coagulans*, *B. subtilis*, *B. cereus*) whose health-promoting properties find applications in both human and veterinary fields [216,217,218].

On the other hand, probiotics also include the potential risk of introduction and/or transfer AMR traits through a range of mechanisms and may also trigger non-genetically determined resistance in the endogenous microflora (phenotypic susceptibility due to the probiotic strain) [167,219]. Lactobacilli have been identified as carriers of vancomycin resistance. Resistance to macrolide, lincosamide, and streptogramin antibiotics is quite often found in lactobacilli, although deepened tailored studies on single probiotic strains are needed. *Bifidobacteria* are reported to carry the *tetW* gene, and the consequent resistance to tetracycline. Notably, β-lactam resistance in *Bifidobacteria* is rare. Nevertheless, defining a generalized trend is rather difficult owing to the important differences observed in the resistance profiles of the bacteria and probiotics isolated from diverse geographical areas [219]. Several research groups and regulatory agencies are focusing on overcoming these issues in order to define standardized guidelines in the assessment of the safety of probiotics before licensing their use in clinical practice [219].

### 5.6. Utility of Monoclonal Antibodies against Pathogens

Monoclonal antibody (mAb) therapy is progressively gaining interest in treating infectious diseases. Indeed, mAbs are an important pharmacotherapy tool with a serious level of particularity. They have unrivalled viability and tolerance when compared with ordinary polyclonal antisera. Generally, mAbs developed for bacterial infections typically target surface-exposed antigens or secreted toxins that are not currently targeted by antibiotics and unlikely to be affected by existing resistance mechanisms [220]. In the race to address the global threat of antibiotic resistance, attention is focused on developing therapeutic antibodies as an alternative approach as it has potential advantages over broad spectrum antibiotics [221]. The history of using antibodies for treating infections in humans dates back to the early 1900s. Antibiotics were soon preferred due to allergic reactions, variable efficacy between lots, and limited spectrum [222]. The advent of molecular biology tools has led to the development of therapeutic mAbs with improved efficacy, safety, and purity, which enabled the successful translation of antibodies to the clinic [220,223]. Most of the currently practiced antibody therapies are aimed at treating diseases of non-infectious origin and only a few antibodies were approved for treating bacterial infections [224].

According to the literature, initiatives to produce mAbs to treat infections caused by nosocomial bacterial pathogens have met with varying degrees of success, and there are now 14 therapeutic mAb products in various phases of development [221]. Raxibacumab, obiltoxaximab, and bezlotoxumab are the three therapeutic mAbs currently licensed for the prevention or treatment of bacterial infections caused by *Bacillus anthracis* and *Clostridium difficile*, respectively [225,226]. There are additionally several mAbs in various stages of development for viral and bacterial conditions. As contrasting options to the customary antivirals and antibacterials, the antimicrobial mAbs are of great importance. These mAbs are more applicable to the administration of conditions, such as arising viral flare-ups where there is an absence of prophylactic antibodies.

### 5.7. Bacteriophages Based-Specific or Selective or Both!

Bacteriophages, also known informally as phages, are viruses that infect, and replicate within, bacteria and archaea. The relationship between the two organisms is parasitic, where the bacteriophages use the biosynthetic machinery of the host to create important components, such as proteins and lipids essential for phage capsid formation and reproduction [227]. Despite appearances, these organisms are not pathogenic to humans or animals and can be used to treat bacterial diseases. Phages exist in two main cycles. In the ‘lytic cycle’, the viral activity within the bacterial cell leads to cellular destruction, while, in the ‘lysogenic cycle’, the genetic material of the phage is inserted into the host bacterial DNA. The lytic phage is the most relevant cycle to the therapeutic treatment of AMR disease [228].

#### 5.7.1. Phage Therapy

The use of phage therapy for bacterial infectious diseases has existed for decades, with bacteriophages being first discovered by Frederick Twort in 1915 and Félix d’Hérelle in 1917, who described a bactericidal effect after isolation from the feces of patients recovering from dysentery [229,230]. During this time, there was a rapid spike in the development of phage therapy, particularly in Europe, the United States, and the Soviet Union [231]. However, this was quickly followed by a decline in phage therapy use during the 1940s, when the production and use of antibiotics increased in the United States as penicillin use took hold and sulfonamide use continued (synthetic antibiotics that were initially discovered in Germany during the 1930s) [231]. Despite this preference, phage therapy remained (and remains) in use in several countries previously allied to the Soviet Union, i.e., Poland, Georgia, and Russia itself [232]. Further, the rise of antibiotic resistance means that there is a potential niche for phage therapy in treating/preventing AMR bacterial infections. For example, phage therapy cured a patient with cystic fibrosis who was infected with disseminated drug-resistant *M. abscessus*, while another patient, infected with a resistant *A. baumannii*, was cured [228,233]. Other examples include the use of bacteriophages in the treatment of colistin-only-sensitive *Pseudomonas aeruginosa* septicemia and burn wound infections, as well as their potential use in elderly patients presenting with relapsing *S. aureus* prosthetic-joint infection [234,235,236]. Additionally, the potential efficacy of phage therapy has been widely explored in farm animals, poultry, and pet animals, especially for zoonoses and animal diseases linked to economic loss, with some encouraging results [237].

#### 5.7.2. Phage-Derived Lytic Proteins as a Antibacterials

Phages that target bacteria encode several lytic proteins, most notably peptidoglycan hydrolases (PGH), called endolysins and virion-associated peptidoglycan hydrolase (VAPGH), necessary for the disruption of phage-infected bacteria, thereby allowing the release of progeny phage particles into the environment. On the other hand, small lytic phages of family *Microviridae* and *Leviviridae* accomplish the lysis of the bacterial host by a single gene, encoding a protein lacking any peptidoglycan degrading activity [238]. Endolysin accumulates in the cytoplasm of the bacterium at the end of the phage replication cycle. The accumulated endolysins can infiltrate the cytoplasmic membrane through holes created by holins and break down extracellular peptidoglycan, allowing the cell to osmolyse.

Currently, phage-derived lytic proteins are being investigated as potential treatments for AMR bacterial infections, food preservation, animal feed, and plant cultivation. For example, purified pneumococcal bacteriophage lytic enzyme (Pal) has been shown to be effective against penicillin-resistant strains of *S. pneumoniae* in the human oropharynx with minimal effect on the commensal flora [239]. Additionally, in vivo investigations carried out on the activity of lysozymes *Cpl-1* and *Cpl-7* reported a significant reduction in the colonization of *S. pneumoniae* in a nasopharyngeal mucosal tissue in a mouse model of infection. Studies have also demonstrated synergistic activities of these endolysins when used in combination with other antibacterial agents [240]. The phage endolysin SAL-1 has been shown to be a novel antibacterial drug for the control of multi-drug-resistant *Staphylococci* infections [241,242]. This was further formulated to N-Rephasin^®^ SAL200 for clinical use. Currently, this formulation is being investigated for safety and efficacy of a single intravenous dose of N-Rephasin^®^ SAL200 for persistent *S. aureus* bacteremia under phase IIa clinical study [243]. Considering the efficacy of phages and phage-derived lysins, the USFDA has approved the application of bacteriophage-based products as an eco-friendly approach for the control of food borne pathogens [240,244]. Further, phage-derived lytic proteins possess several advantages over antibiotics, namely: (a) rapid and extensive bactericidal action against the target pathogen, (b) lack of resistance development due to their specific action on conserved structural components of bacteria, (c) synergistic action with other lysins or antibiotics, and (d) their effect on phenotypically resistant persister cells growing on mucosal surfaces or in biofilms [245]. Continuing advances in genetic engineering have further revolutionized the area of bacteriophage research and identified and characterized several endolysins active against a range of pathogenic bacteria [240,246].

### 5.8. Biofilm Dispersion Methods

Biofilms are microbial communities that exist as complex structures that may be present on biotic or abiotic surfaces in food and medical sectors. Biofilm formation is an adaptation and survival strategy and is frequently the underlying reason for the failure of antibiotic treatment, responsible for 65 to 80% of all infections [247]. The biofilm develops as a multi-layer complex community that may be mono- or polymicrobial in nature [248]. Biofilms are initially formed via reversible attachment of single cells of “planktonic phase“ bacteria to surfaces, with most bacterial cells subsequently becoming encased in a self-produced extracellular matrix material, referred to as “sessile communities”. Bacterial cells may continually escape (’disperse’) from biofilms, particularly during the later stage of biofilm formation. This dispersion process is a possible mechanism for reducing biofilm mass since the free-living ‘planktonic’ phase of the bacterial lifecycle is often more sensitive to antimicrobial drugs and immunological responses. [249]. In fact, biofilm dispersion mechanisms can be split into two categories: active and passive dispersion. The production of enzymes that destroy the biofilm matrix and promote dispersion is dependent on a reduction in intracellular c-di-GMP levels, which leads to *active* dispersion. On the other hand, *passive* dispersion is based on triggers that directly release cells from the biofilm [250]. In this respect, both intercellular quorum sensing signals and intracellular c-di-GMP signaling are involved in biofilm development. Interference of these two signaling pathways is being used to develop new biofilm control approaches [251]. However, both active and passive dispersion strategies have their drawbacks. The transcriptome of experimental model dispersed cells is distinct from that of biofilm and naturally released planktonic cells, with dispersed cells possessing lower c-di-GMP concentrations—associated with greater virulence and implying that dispersed cells are more pathogenic than planktonic cells. The timing and concentration of dispersion treatments also remains a challenge [250].

### 5.9. Discovery and Role of Anti-Persister Antimicrobials

Persistence was discovered in 1944 by Joseph Bigger and involved colonies of bacterial cells that could not be completely eliminated by penicillin treatment. This residual population of bacteria remained viable after antibiotic exposure. These populations of cells were not antibiotic-resistant mutants but rather bacterial subpopulations that could resist antibiotic treatment by physiological adaptation [252]. Bacterial persistence is associated with an increased risk of AMR during treatment as persister cells, which have the ability to cause recolonization and relapse after antibiotic treatment, ultimately lead to chronic resistant infections [253]. Anti-persister drugs have been developed to enhance the eradication of persister cells. A classic example is the combination of tobramycin (an aminoglycoside antibiotic) with fumerate (antipersistance compound) to reduce chronic *Pseudomonas aureus* infections [254]. An ideal anti-persister compound should be able to passively transport inside the bacterial cell and independently kill persister cells without the requirement of active metabolism [255]. The identification of novel molecules with anti-persister activity can be one of the strategies to tackle AMR. Future anti-persister approaches should also target the membrane structure with enhanced permeability for slow growing pathogens [255].

### 5.10. Disruption of Quorum Sensing

The processes of biofilm formation and quorum sensing (QS) are inextricably linked. Biofilm development is cooperative group behavior in which bacterial populations live immersed in an extracellular matrix that they produce themselves. QS can be defined as a cell–cell communication process that harmonizes and regulates gene expression [256]. QS is different for Gram-negative and Gram-positive bacteria [257]. Bacteria use signaling molecules, called autoinducers (Ais), to synchronize gene expression, virulence, and biofilm formation. On the other hand, bacteria, respectively, use quorum sensing inhibitors and quorum quenching enzymes to control AIs and to degrade signaling molecules [258]. The microbial quorum induction can be controlled by the use of quorum-quenching agents, leading to the reduction in microbial infections, pathogenicity, biofilm formation, and also to increasing bacterial susceptibility to antimicrobial agents, such as antibiotics and bacteriophages [258,259]. For example, a study employing mathematical modelling was conducted to assess the effect of combining the therapy strategies of a quorum-quenching enzyme and a quorum-sensing inhibitor in controlling quorum sensing pathways in *P. aeruginosa*. These results were promising when used in vitro, and further research is needed to focus on determining the efficacy of the combined therapy in vivo [260]. The identification of potential new research is important before the use of QS-based treatments against pathogens [261].

## 6. Strategies Based on Drug Delivery Systems

### 6.1. Facilitated Drug Delivery Systems

One of the main challenges that antibiotic research is facing is the poor cell permeability of antibiotics. The development of antimicrobial delivery systems is a promising approach to enhance the entrance of the antibiotic into the intracellular space [262]. An important strategy is to take advantage of bacterial iron transport systems. Synthetic siderophores analogues can act as a delivery system when conjugated with antibiotics, facilitating their entrance into the bacteria. The inclusion of a conjugate with ampicillin was reported to have a 1000-fold increase in activity against *P. aeruginosa* and 100-fold increase in the case of gram-negative enterobacteria. They also demonstrated that the conjugate siderophore + ampicillin was not a substrate for efflux pumps in *P. aeruginosa* and, therefore, was able to evade one of the main resistance mechanism in this species [263]. Another example of conjugates with antipseudomonal activity was with the use of an artificial tris-catecholate siderophore with a tripodal backbone to subsequently conjugate it with ampicillin and amoxicillin. Both conjugates considerably enhanced the in vitro antimicrobial activity of the antibiotics by using energy-dependent iron uptake systems to cross the outer membrane [264].

On the other hand, polymeric nanoparticles are emerging as a strategy to improve the solubility, stability, and bioavailability of antimicrobials as well as reduce the exposure of the microbiota to sub-lethal doses that can lead to the development of resistance. Polymeric nanostructured systems allow precise drug release based on different methods, such as diffusion, elution, or chemically/stimuli-controlled, and they are synthesized from natural precursors, such as chitosan, collagen, or gelatine, which makes them biocompatible and less toxic. Synthetic precursors, such as polylactic acid or polyethylene glycol, can also be found [265,266]. Developed phosphatidylcholine–chitosan hybrid nanoparticles coated with gentamycin confirmed the capacity of the construct to avoid biofilm formation and bacterial growth in both Gram-negative and -positive bacteria [267]. Another variant nanocapsule demonstrated that the hydrophilic polymersomes encapsulated vancomycin, which also serves to treat infections more efficiently, in this case, those caused by methicillin-resistant *S. aureus* [266].

Since 1990, biodegradable nanoparticles such as nanocarriers have been studied to improve drug delivery. Nanoencapsulation has proved to increase antimicrobial efficacy and efficiency by protecting it from degradation, enhancing the targeting accuracy, and increasing cellular uptake [268,269]. A typical approach to generate nanocarriers involves nanovesicles, for instance, metallic and, as mentioned above, polymeric nanoparticles, liposomes, carbon nanotubes, or dendrimers. Liposomes, small lipid-based nano-systems composed of a concentric phospholipid bilayer and a biodegradable structure, are versatile in their delivery of both hydrophobic and hydrophilic drugs due to their external liposomal bilayer and the posterior internal aqueous compartment [270,271]. Some liposomes are already available in the market as topical formulations, such as polyvinyl-pyrrolidone-iodine hydrogel, with activity against a wide range of bacteria for use on external wounds [272]. This strategy provides the possibility of regulating the dose at higher than MIC concentrations while reducing dose-dependent toxic effects [271]. Furthermore, it was demonstrated that liposomes increased the antimicrobial activity against species such as *E. coli*, *K. pneumoniae*, *A. baumannii*, or *P. aeruginosa* by entrapping polymyxin B in DPPC/Chol-1,2-dipalmitoyl-sn-glycero-3-phosphocholine and cholesterol- liposomes [273].

This same concept of conjugation is being studied with monoclonal antibodies. Antibody–drug conjugates consist of a monoclonal antibody that binds, through a covalent bond, with a chemical drug, such as an antibiotic, although it has wider use in antitumoral drugs. An antibody–drug conjugate was approved for the first time by the FDA in 2000 (gemtuzumab–ozogamicin) used in the treatment of myeloid leukemia [274]. For bacterial infections, the molecule DSTA4637S (a novel THIOMABTM IgG antibody linked by a protease cleavable linker to rifamycin class antibiotic (dmDNA31)) conjugate) is an antibody conjugate currently studied at preclinical levels as a potential treatment for complicated *S. aureus* bacteriemia [275].

### 6.2. Anti-Plasmid and Plasmid Curing-However, Not Suitable for In Vivo

Plasmids can confer resistance to almost all classes of antibiotics. In fact, the dissemination of AMR genes among Gram-negative bacteria is importantly attributed to plasmid mobilization by conjugation, transformation, or transduction [276]. Novel anti-plasmid and plasmid curing strategies are intended to reduce the prevalence and spread of AMR genes. In other words, plasmid curing procedures are performed to remove plasmids from bacterial populations, which is a striking approach to combatting AMR. Unfortunately, few curing mechanisms have been tested “in vivo”. Research in this area is greatly needed because of the potential for curing agents for humans and animals, especially food-producing animals. Other approaches, with a One Health perspective, consider curing plasmids in environmental hot spots, such as wastewater or agricultural settings [277].

However, despite applicability issues, anti-plasmid technology is successfully progressing. Curing agents encompass chemicals to more sophisticated genetic engineering tools, such as CRISPR-Cas Systems. Chemical compounds targeting plasmids include detergents, DNA intercalating agents, or psychotropic drugs. Bile acid detergents can lead to the loss of the *Salmonella Typhimurium* plasmid *pSLT* [278], or that the heterocyclic compound phenothiazine is effective in curing plasmids from *E. coli* [279]. Chlorpromazine, for example, can eliminate plasmids from *P. aeruginosa*, *P. mirabilis*, or *E. cloacae*, and the DNA intercalating agent ethidium bromide has been known to cure plasmids in *S. aureus* since the early 1970s [280]. DNA intercalators nonetheless have a powerful mutagenic activity, leading to significant toxicity and carcinogenic consequences that confine its use to “in vitro” plasmid curing.

Alternatively, apart from chemical strategies, the principle of plasmid incompatibility can be exploited to eliminate plasmids. Plasmids belonging to the same incompatibility groups cannot survive inside a cell as they compete for the same resources. In this way, introducing small high copy-number plasmids will eliminate a resident plasmid. This method can reduce associated toxicity to curing agents and avoid the minimum chromosomal mutations. An example of curing by incompatibility is constructing the *pCURE1* plasmid, designed to target pO157, an *IncF* plasmid present in the host *E. coli* O157:H7 [281]. The possibility of delivering plasmids to humans or animals through bacteria or phages is currently under study, with the main handicap being antimicrobial pressure and resistance cassettes in selecting curing plasmids. In the case of bacteriophages, sometimes merely their presence is sufficient to cure a plasmid, such as the presence of the filamentous phage M13, whose minor coat protein g3p is not only necessary but enough to avoid F-plasmid conjugation in *E. coli* strains [282].

A novel approach includes the use of the CRISPR-Cas systems to target AMR genes present in plasmids and, sometimes, cure the plasmid completely due to the destabilizing double-stranded breaks generated by the nucleases [148,283]. This system can be adapted in transformative and conjugative plasmids. For example, the gene *mcr-1* confers resistance to colistin and subsequently introduced colistin-resistant plasmids to an *E. coli* strain. A transformative plasmid followed by a conjugative plasmid containing the CRISPR-Cas9 targeting the gene *mcr-1* was demonstrated to not only interrupt the *mcr-1* gene but to cure the plasmid completely out of the cell and, in this way, eliminate the resistance to colistin [284]. Although the utility of the system to resensitize bacteria is undeniable, the delivery method is still a barrier, especially for “in vivo” models. Bacteriophages and phagemids have also been studied to deliver plasmids incorporating the CRISPR-Cas system on them, which resulted in good outcomes “in vitro” but, again, limited when applied “in vivo” [283,285].

### 6.3. Antivirulence Compounds

Another approach different from typical cell growth inhibitors is reducing the pathogenicity of clinically relevant species through the development of antivirulence compounds. This strategy prevents the development of resistances by attacking pathogenesis pathways but does not affect bacterial viability and, therefore, helps to reduce their spread, while the host immune system proceeds with bacterial clearance. Numerous factors are involved in bacterial virulence, so, depending on the target of the compound, there are different categories of virulence inhibitors [189].

Quorum sensing inhibitors affect the bacterial communication system, often required to modulate virulence responses, which makes them ideal to reduce the pathogenicity of the species in question. Quorum sensing regulates actions such as siderophore production, biofilm formation, or protease releasing. Interruption of quorum signaling can occur through synthase inhibitors that block the synthesis of signal molecules. Other stages of the pathway can be targeted. Quorum signaling can be inhibited by specific enzymes that degrade the signaling molecules. Environmental conditions can also affect this stage of the process. Additionally, analogs/antagonists of signaling molecules can attach the receptor of the molecules involved in quorum sensing, displacing them and stopping the process [286]. Quorum sensing inhibitors can be natural and synthetic. Flavonoids are natural plant metabolites known to target the autoinducer-binding receptors, *LasR* and *RhlR*, essential for quorum sensing, in *P. aeruginosa*. The presence of the two hydroxyl moieties in the A-ring backbone of the flavone is fundamental for its antagonistic activity [287,288]. Quenching acyl-homoserine lactone with lactonases is also an effective strategy against *P. aeruginosa* [289]. In another study, the potential activity of synthetic derivates of 3-acylpyrrole, which is a motif present in many drugs and biologically active compounds, to control *V. cholerae* infections through quorum sensing inhibition was examined [290].

Antivirulence compounds can also act at numerous levels, as, for example, host immune modulation, such as Lipid A inhibitors [189]. For example, fimbria antagonists and pili formator inhibitors target secretion systems or two-component systems. *E. coli*’s *FimH* is a Type 1 fimbrin D-mannose adhesin precursor and is one of the best known carbohydrate-specific lectins, and α-D-mannosides are a promising approach to inhibit fimbria formation through their *FimH* antagonism capacity [291]. Another virulence factor, especially important in Gram-negative bacteria, is Type Three Secretion Systems (T3SSs), which allows pathogens to inject virulence proteins directly into host cells, easing disease progression. It was demonstrated that the salicylidene acylhydrazydes were used to block the virulence of *S. enterica* serovar *Typhimurium* [292,293].

## 7. Physicochemical Methods

### 7.1. Atmospheric Pressure Non-Thermal Plasma (APNTP)

The non-thermal (or cold) atmospheric pressure plasma is an emerging technology that is currently under investigation concerning antimicrobial properties [294]. APNTP in vitro application for a range of microorganisms indicated that it is effective in the inactivation of bacteria, viruses, fungi, and parasites. Due to its relative ease of use and cost-effective operation with only limited local side effects known to date, it presents a promising potential alternative to conventional antimicrobial treatments, including for some infections in certain applications [295]. This field of plasma medicine has been evaluated for bacterial inactivation, air sterilization, tooth root canal therapy, and wound healing, especially where traditional antibiotics often fail [296]. The exact mechanisms of APNTP-mediated bacterial inactivation are still under investigation, but it seems to be effective through generated products, such as Reactive Oxygen Species (ROS), Reactive Nitrogen Species (RNS), UV radiation, and charged particles within a plasma gas phase [297]. The ROS considered to be involved in bacterial inactivation are ozone, atomic oxygen, singlet oxygen, superoxide, peroxide, and hydroxyl radicals [298,299]. The basic operation of these mechanisms is the damage of nucleic acids by UV radiation, lipid peroxidation caused by ROS occurring mainly in fatty acids near the cell surface, and the chemical modification and degradation of proteins caused mainly by hydroxyl radicals. Other studies also reported apoptosis in bacterial cells probably induced by ROS [300]. Mechanical cell damage, in particular electrostatic disruption caused by the electrostatic forces of charged particles, accumulates on the cells. Mechanical cell damage is also caused by electroporation through the direct bombardment of charged particles [301].

Interestingly, it was demonstrated that Gram-negative organisms were more sensitive to APNTP than Gram-positive organisms, indicating that APNTP-induced damage to the cell membrane and cell wall is a critical factor [295]. Clinically important bacteria have been reportedly inactivated using cold plasma [302]. APNTP was also shown to inhibit bacteria both in suspension and in biofilms [294]. The in vitro inactivation of *P. aeruginosa* biofilm was also demonstrated using the cold plasma system [298]. A potential advantage of APNTP is the inactivation of bacteria without damaging mammalian cells [295,303,304]. Cold plasma can also be combined with antibiotic therapy, although attempts to quantify the cold plasma dosage are yet to be found [305].

### 7.2. Sonodynamic Antimicrobial Chemotherapy

Sonodynamic antimicrobial chemotherapy (SACT) is based on the synergistic effect of ultrasound (US) and a chemical compound referred to as “sonosensitizer” (SS) [306], whereby an inaudible sound with a frequency less than 20 kHz is capable of killing microorganisms [307,308,309]. SACT uses the sensitization of the target site with a non-toxic sonosensitizer, relatively low-intensity US, and molecular oxygen, which may produce micro-bubbles through the acoustic cavitation process during the interactions between the US wave and target cells [13]. Inactivation of *E. coli* was reported by the application of US in combination with the conventional antibiotics Ciprofloxacin and Levofloxacin due to enhanced uptake and the production of cytotoxic ROS [310,311]. However, the ability of US to enhance the bacterial uptake of antibiotics without activation by radiation is well established [312]. More importantly, due to the excellent regional focusing characteristics and the ability to penetrate strong tissue, SACT is known to be a more efficient therapy with fewer side effects [13]. The advantage of US from a clinical point of view is a very good tissue penetrating ability without major attenuation of its energy. This is an attractive feature, and an extensive evaluation is warranted [313].

Both organic and inorganic sonosensitizers have been identified. Many inorganic sonosensitizers have superior physiochemical properties, but the clinical translation remains unresolved because of non-biodegradation and potential biosafety issues. However, organic sonosensitizers have the advantages of clear structure and easy metabolism, which is conducive to clinical applications [314].

### 7.3. Photoinactivation

Photoinactivation or photodynamic antimicrobial chemotherapy (PACT) is a promising strategy to eliminate pathogenic bacteria, which utilizes visible light, a photosensitizer (PS)-chromophore, and molecular oxygen to create reactive oxygen species (ROS), resulting in bacterial cell death [315,316]. PACT has been demonstrated to act on a wide range of bacteria, i.e., Gram-negative and Gram-positive, antibiotic-resistant, or susceptible bacteria strains [315]. This technique has gained much research attention as an alternative strategy to combat AMR [317,318]. The advantages of PACT over the conventional antibiotics in clinical settings include its localized wound application and minimal side effects, resistance, and toxicity (Huang et al., 2010). In contrast to antibiotics, sub-inhibitory doses of photodynamic inactivation (PDI) have failed to induce genomic mutations and elevate antibiotic or photodynamic resistance [319,320].

Several improvements have been made to position PACT as an effective alternative to antimicrobial chemotherapy, for example. Antimicrobial inactivation is the conjugation of porphyrins to nanoparticles [315]. By taking advantage of the small size of the porphyrins and porphyrin-nanoparticle conjugates, the photosensitizers can attach to the bacterial cell wall through a self-assembly process, resulting in cell deaths [315].

It is pertinent to improve the application of PACT in dermatological and control of infectious diseases, especially in the management of acne and skin infections in general. Given the wide range of bacteria, it has been noted that the potential application of PACT in the treatment of infectious diseases is still lagging [315]. While the application of the PACT systems has been extensively evaluated for the topical/local approach for animal model evaluations, more studies on systemic application still need to be done to fully evaluate their in vivo stability and therapeutic modality. It is important to fully understand their mechanisms of action and fine-tune them appropriately to improve their sensitivity and selectivity. Notably, most studies lacked toxicity data, and there is a need, therefore, for future studies to include toxicity studies. Toxicity profile evaluation will be important in providing confidence in PACT systems before submitting the final products for regulatory endorsements [315].

The PACT is also severely limited by the inability of light to penetrate to depth through mammalian tissue, mainly due to endogenous pigments, such as melanin, competing for light absorption with the sensitizer, and it is a particular problem in localized infection where the wound area may be severely discolored due to bruising or inflammation, or in ethnic groups where the skin is naturally heavily pigmented (Huang et al., 2008). Currently, approved sensitizers absorb in the visible region of the electromagnetic spectrum, limiting light penetration to only a few millimeters and reducing the ability of APDT to eradicate bacteria located deeper within infected wounds [321].

### 7.4. Other Physicochemical Means

Metals and metal oxides have been widely studied for their antimicrobial activities [322]. Metal oxide nanoparticles, well known for their highly potent antibacterial effect, include silver (Ag), iron oxide (Fe_3_O_4_), titanium oxide (TiO_2_), copper oxide (CuO), and zinc oxide (ZnO). Most metal oxide nanoparticles exhibit bactericidal properties through the generation of ROS, although some are only effective due to their physical structure and metal ion release.

Following the European Union (EU)-wide ban on the use of antibiotics as growth promoters, the use of copper or zinc was promoted in food animals and aquaculture. However, the use of copper or zinc results in not only damage to the environment but also might promote the spread of antibiotic resistance via co-selection [323,324]. It is now under legislation to ban the use of ban ZnO as a veterinary medicinal product above the levels of 150 ppm in the EU from June 2022 [325].

## 8. Expected Role of Vaccines in Combating AMR Pathogens

Vaccination represents an interesting approach to prevent the development of an infection and/or disease in humans and animals, reducing the use of antibiotics and, thus, preventing the emergence and spread of resistant pathogens. Pneumococcal conjugate vaccines are a clear example that vaccines are effective when it comes to reducing AMR, and so is the *H. influenzae* type B vaccine. In the 1990s, before the polyvalent pneumococcal conjugate vaccine, there were approximately 63000 cases of pneumococcal disease in the US, leading to an important increase of resistance to penicillin in *S. pneumoniae* among other classes of antibiotics. After the introduction of the vaccine, it not only reduced the prevalence of the disease but significantly reduced the bacterial colonization of this pathogen, which clearly affected the spread of AMR-strains [326].

The role of vaccines in fighting AMR is both direct and indirect. Vaccines have a direct effect on resistant pathogens by reducing the infection incidence and, indirectly, through reducing the circulation of AMR-resistant strains to other non-resistant species. A reduced incidence of infections is linked with a diminished prescription of antibiotics and a reduced onset of secondary infections and superinfections that would, otherwise, unavoidably require massive antibiotic usage [327,328]. The reduced circulation of resistant pathogens makes vaccinal strategy and the achievement of herd immunity, a valuable tool against AMR [329]. Recently, vaccination programs in livestock and poultry resulted in a drastic reduction of antibiotic usage and a concomitant decrease of AMR in the herd [330], with important benefits also to human health through reduced circulation of AMR traits in zoonotic agents [328]. Therefore, vaccination leads to decreased load of AMR and, hence, the use of antibiotics; otherwise, the overuse of antibiotics represents a major driver for the emergence and spread AMR in veterinary, agriculture and aquaculture.

Furthermore, vaccine strategies over the past decades enabled the eradication of important pathogens, such as smallpox and rinderpest virus [330]. Vaccines are also responsible for the near complete elimination of poliomyelitis, diphtheria, tetanus, pertussis, measles, mumps, and rubella [331]. Moreover, the mechanism by which vaccines reduce pathogens and AMR circulation is completely different from those of antibiotics, resulting in little to no selective pressure on microorganisms; thus, the probability of the emergence of resistant pathogens is much lower than for antibiotics.

Resistance to antibiotics is a common occurrence, while vaccination resistance is relatively minimal [332]. Vaccine resistance has been documented for some critical pathogens, such as hepatitis B virus [333] and *Bordetella pertussis* [334]. Moreover, vaccines commonly target a plurality of antigens, raising the likelihood of success of the prophylactic measures over time; whereas, antibiotic molecules concern a single target, resulting in a drastic loss of efficiency once a single mutation occurs in the microbial specimens [328,335].

Nevertheless, the development of vaccines against the major antibiotic-resistant pathogens is still needed, and important efforts are being undertaken in this area. Recent advances in the research of omics sciences and bioinformatics open new avenues for innovative vaccines against pathogens, including antibiotic-resistant ones. Traditional vaccinal strategies rely on the administration of live attenuated pathogens or inactivated germs. Both methods enable efficient immunization and the stimulation of a good immunological memory, although the second strategy is safer and less efficient and multiple stimulations are necessary to maintain the protection of the vaccinated subjects. Following the advent of the recombinant DNA techniques, vaccinology has powerfully evolved with the production of recombinant vaccines featuring higher safety levels and the immunological power of the live attenuated vaccines. This ensures better control of pathogens, such as hepatitis B virus, and the generation of the genetically detoxified *B. pertussis* toxin [336]. Recombinant DNA technology has been employed to produce subunit vaccines, where is it expected that the administration of antigenic determinants only will provide good protection and almost no risk to the vaccinated subject’s health. This technology has been successfully applied to produce the influenza attenuated vaccine and the rotavirus vaccine [336,337,338].

Among the latest achievements in the field of vaccinology, reverse vaccinology is also worthy of note. This innovative approach relies on the genomic information of the pathogen of interest, namely the list of antigens and epitopes the microorganism is capable of expressing, thus enabling the rapid and fair screening of candidate antigens to be employed for the vaccinal formulation [339]. Reverse vaccinology has led to an efficient vaccine against meningococcal serogroup B; 4CMenB was reported. A similar approach has been pivotal for the selection of antigen candidates for the development of a vaccine against *E. coli* [340] and *P. aeruginosa* [341].

Another innovative vaccinology strategy is the use of outer membrane vesicles as a vaccine platform. This approach relies on the production of outer membrane-based vesicles from Gram-negative bacteria opportunely modified to reduce lipopolysaccharide-mediated reactogenicity and other unwanted interfering reactions. These serve as safe and potent immunostimulants against Gram-negative bacterial pathogens, such as *Neisseria meningitidis* and *Shigella sonnei* [342,343,344,345].

Recently, the generation of antimicrobial glycoconjugate vaccines has been suggested as a simple in vivo strategy to produce vaccines against antibiotic-resistant bacteria. This simple strategy relies on the genetic modification of a single strain of *E. coli* to produce the desired vaccine through oligosaccharyltransferase, PglB. This enzyme catalyzes the linkage of a selected antigenic polysaccharide to target carrier proteins that contain specific N-glycosylation sites. Through the guided glycosylation of specific polysaccharide bacterial antigens, it is possible for the efficient production of glycoconjugate vaccines against antimicrobial-resistant pathogens, such as *Shigella flexneri, S. aureus, P. aeruginosa*, and *K. pneumoniae* [346,347], in a safe (there is no requirement to grow these high-risk pathogens) and efficient way [348].

A recent approach for combining antibiotics and vaccines to treat AMR is the use of a hypothetical narrow-spectrum vaccine to target the most resistant strains, thereby encouraging replacement with susceptible-strains. Antibiotics might then be used to successfully treat these infections [332].

Finally, there is a theoretical possibility that reducing the density of microbial populations by vaccination reduces the opportunities for genetic exchange of resistance elements for vaccines against organisms, such as *S. pneumoniae*, *S. aureus*, and members of the Enterobacterales family, which asymptomatically colonize the nasopharynx, skin, gut, and other sites [349].

While vaccines are not meant to replace antibiotics, they can help to minimize AMR by reducing the antimicrobial use (AMU), to avoid illness resulting from AMR bacterial infections, and to prevent the spread of AMR bacteria.

## 9. Unconventional Strategies

In view of the AMR emergency, several alternative drugs have been tested, which include antihistaminic, anesthetics, non-steroidal anti-inflammatory drugs (NSAID), antipsychotics, and cardiovascular drugs [350]. Fecal microbiota transplantation (FMT) is an important alternative strategy being tested to fight AMR. Through the administration of fresh, frozen, or encapsulated fecal matter from a suitable donor, the unhealthy gut microbiota of the patient is restored, re-establishing the alpha-diversity. Treatment with FMT is especially recommended for recurrent *Clostridium difficile* infections, showing over a 90% of efficacy in randomized clinical trials [351]. FMT is also efficient in displacing vancomycin-resistant Enterococcus when they are predominant over the rest of gut microbiota and also when *C. difficile* is present [352].

## 10. Conclusions

AMR, especially antibiotic resistance, continues to emerge and spread beyond all boundaries. This is not a single issue; rather, it is associated with multiple parameters. Coordinated efforts and multidisciplinary collaborations are required to tackle AMR at the local, national, and international levels. Strong political commitment can play a vital role in formulating policy, implementation, and regular educational updates based on scientific evidence to better regulate the use and sale of antibiotics for both humans and animals. The unethical promotion of antibiotics must be controlled, and strategies to eliminate the overuse or inappropriate use of antibiotics must be implemented. Several new approaches have been tested to enhance antibiotic efficacy through novel targets and mechanisms that include resistant gene inactivation, silencing, and editing. Importantly, most of the novel alternative strategies do not trigger antibiotic resistance. Fortunately, many new avenues are being explored with the view to combatting current and emerging resistance, although it will be a number of years before we will be able to determine their utility—both alone and in combination—with efforts being made by regulatory authorities, institutions, and governments.

## Figures and Tables

**Figure 1 antibiotics-11-00200-f001:**
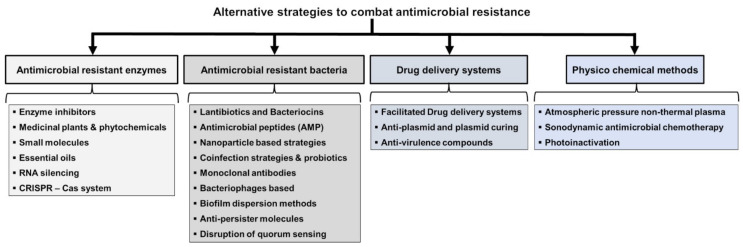
Categories of alternative strategies to combat antimicrobial resistance.

**Figure 2 antibiotics-11-00200-f002:**
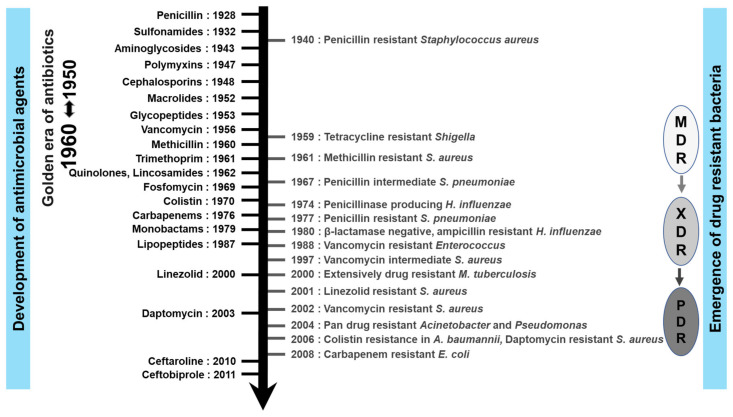
Timeline of eight decades of antimicrobials discovery alongside AMR emergence. MDR: multi-drug-resistant, XDR: extensively-drug-resistant, and PDR: pan-drug-resistant.

**Figure 3 antibiotics-11-00200-f003:**
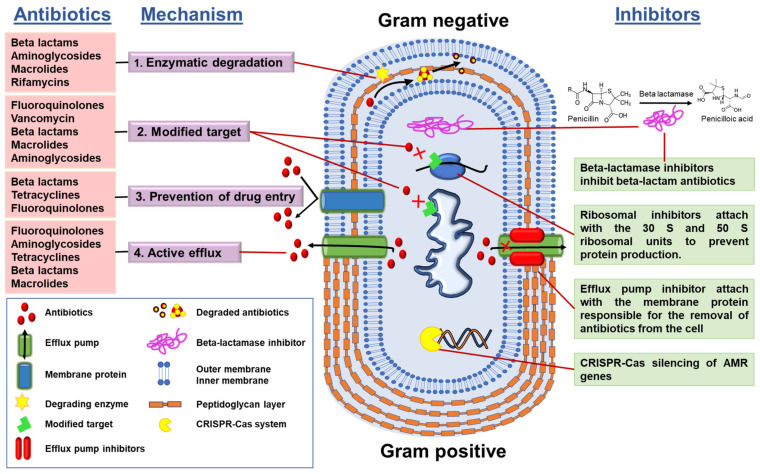
Classes of antibiotics, mode of action, and inhibitors.

**Table 1 antibiotics-11-00200-t001:** Antibiotics class and mode of action.

Antibiotic Class	Mechanism of Action	References
Beta lactams: carbapenems, cephalosporins, monobactam, penicillin, glycopeptides	Inhibit cell wall synthesis	[15]
Lipopeptides	Depolarize cell membrane	[16]
Aminoglycosides, tetracyclines Chloramphenicol, macrolides	Inhibit protein synthesis by binding to 30S ribosomal unit and 50S ribosomal unit	[17,18]
Quinolones	Inhibit nucleic acid synthesis	[19]
Sulfonamides, trimethoprim	Inhibit metabolic pathways	[20]

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
