# Peer review of "Progress in Alternative Strategies to Combat Antimicrobial Resistance: Focus on Antibiotics"

_antibiotics, 2022, doi:10.3390/antibiotics11020200_

Round 1
Reviewer 1 Report
This paper is well written, no significant editorial changes are required
Author Response
We thank the reviewer for time reading our manuscript and for positive decision
Reviewer 2 Report
The manuscript is well focused and organized. This is a topic of great interest, since antibiotic-resistant pathogens are causing many diseases to become persistent and there may be a higher risk of death. These resistant bacteria are a big problem, especially in hospitals, so a review of new therapies to control them is necessary. So I just have a few suggestions to make to the authors.
- I have not found the years covered by the review, I understand that is all that has been found, but until when?
- In figure 1, the last color is very dark and the letters are not well read. I would suggest making it clearer.
- I miss, in general, examples of microorganisms in which they have been tested with good results and efficacy data. For example, in point 3.3. There is talk of therapies but there are no examples of what pathogen has been tested against (only say Gram-negative) and if it has given good results. This would help to get an idea of whether the therapy could really be of interest.
- Review the numbering, which goes from 3.3 to 3.2.1 and from 5.7 to 5.8.1.
- In line 889, reference [329] is of the vaccination of animals for human consumption, a little more reference could be made to the theme of that article of how they indirectly affect the decrease in the use of antibiotics that pass to humans. As in line 268.
Author Response
We thank the reviewer for the positive comments, which have greatly improved the manuscript. The following lists our response to the reviewers comments and suggestion,
- Comment: I have not found the years covered by the review, I understand that is all that has been found, but until when?
Answer: The development happening in the last twenty years were reported. However, the years were not included in the review.
- Comment: In figure 1, the last color is very dark and the letters are not well read. I would suggest making it clearer.
Answer: Now the figure shading is revised as suggested.
- Comment: I miss, in general, examples of microorganisms in which they have been tested with good results and efficacy data. For example, in point 3.3. There is talk of therapies but there are no examples of what pathogen has been tested against (only say Gram-negative) and if it has given good results. This would help to get an idea of whether the therapy could really be of interest.
Answer: Thank you for your comment. We have revised this section to include the examples.
- Comment: Review the numbering, which goes from 3.3 to 3.2.1 and from 5.7 to 5.8.1.
Answer: The section and subsections numbers – corrected in the revised manuscript.
- Comment: In line 889, reference [329] is of the vaccination of animals for human consumption, a little more reference could be made to the theme of that article of how they indirectly affect the decrease in the use of antibiotics that pass to humans. As in line 268.
Answer: Yes, we have included the following in the revised manuscript (line 957-959), “Therefore, vaccination leads to decreased load of AMR and hence the use of antibiotics, otherwise overuse of an-tibiotics represents a major driver for emergence and spread AMR in veterinary, agriculture, and aquaculture” to emphases how the vaccination is beneficial.
Reviewer 3 Report
This article is a review aiming to present alternatives to the use of conventional antimicrobials (AMD) in order to counteract development antimicrobial resistance (AMR) and the lack of AMD efficacy in the event of AMR.
Major comments
This review is clearly written and offers many perspectives. That said, I don't find it very critical and its weakness is to simply list many options without prioritizing them. I believe that the authors should explain in an introductory section that a simple mechanism of action may just opens the way to what is possible, that positive in vitro results may transform the possible into plausible and that only in vivo data (animal model) allow to transform the plausible into probable, which ultimately will have to be confirmed by clinical trials. Indeed, a simple mechanism of action does not necessarily make a drug and even, apparently positive in vitro data, do not necessarily translate into therapeutic advances. As an example, we can cite the case of the penicillin / streptomycin combination, the synergistic character of which has been repeated for decades (including in this review lines 176-179) following the historical work of Jawetz 1953 (Gunnison et al., 1953 ) but whose clinical interest has been disputed (Paul et al., 2004) or at least debated (see the review by ( Tamma et al., 2012) ).
We now know that the drugability of certain approaches such as bacteriophages is very difficult in particular for industrial reasons of manufacture, stability of treatments over time etc. or else, it is necessary to explain why these tracks opened 100 years ago are always promises more than realities apart from a few niche applications.
Another aspect that deserves to be highlighted is the feedback we have today on certain promises made more than 20 years ago, such as that of probiotics. At the time, the promises of probiotic promoters were only positive, but we now know that things are not so simple and that probiotics can promote the spread of AMR (this is indeed reported in this review line 526 but could be used to illustrate the fact that the safety component of new approaches must be taken into account very early in a drug development) . In addition, Authors could point out that everything that is "natural" is not necessarily “safe” . For example, essential oils (section 4.2) can be source of resistance ( Berdejo et al., 2020) and cinnamaldehyde Induces expression of efflux pumps and multidrug resistance in Pseudomonas aeruginosa ( Tetard et al., 2019) . Another aspect which is not addressed at all is environmental toxicology. We know today that the promotion of the use of copper or zinc in veterinary medicine and aquaculture was a very bad idea both for environmental toxicology (Fu et al., 2016) and for the promotion of resistance (Seiler & Berendonk , 2012) what has justified for example, the ban of the pharmacological use of Zno the EU .
One last major comment; I find it surprising that vaccines are viewed as " unconventional strategies ” or else the title of section 9 should begin line 870. Furthermore, it seems to me that their potential role is minimized in this review whereas according to the last seminal review just published in The Lancet , which must be cited and taken into account (Murray and al., 2022) , vaccination is a major strategy to reduce AM.R
Minor comments
-line 90, quote (Murray et al., 2022)
-Several references are incorrectly numbered in the text (example ref 81 line 266 is not related to ceftiofur; line 535, reference 218 is rather 219?
-line 336: merck , not Merch ,
References
Berdejo, D., Pagán, E., Merino, N., Pagán, R., & García-Gonzalo, D. (2020). Incubation with a Complex Orange Essential Oil Leads to Evolved Mutants with Increased Resistance and Tolerance. Pharmaceuticals, 13(9), 239. https://doi.org/10.3390/ph13090239
Fu, Z., Wu, F., Chen, L., Xu, B., Feng, C., Bai, Y., Liao, H., Sun, S., Giesy, J. P., & Guo, W. (2016). Copper and zinc, but not other priority toxic metals, pose risks to native aquatic species in a large urban lake in Eastern China. Environmental Pollution (Barking, Essex: 1987), 219, 1069–1076. https://doi.org/10.1016/j.envpol.2016.09.007
Gunnison, J. B., Shevky, M. C., Bruff, J. A., Coleman, V. R., & Jawetz, E. (1953). Studies on antibiotic synergism and antagonism: The effect in vitro of combinations of antibiotics on bacteria of varying resistance to single antibiotics. Journal of Bacteriology, 66(2), 150–158. https://doi.org/10.1128/jb.66.2.150-158.1953
Murray, C. J., Ikuta, K. S., Sharara, F., Swetschinski, L., Robles Aguilar, G., Gray, A., Han, C., Bisignano, C., Rao, P., Wool, E., Johnson, S. C., Browne, A. J., Chipeta, M. G., Fell, F., Hackett, S., Haines-Woodhouse, G., Kashef Hamadani, B. H., Kumaran, E. A. P., McManigal, B., … Naghavi, M. (2022). Global burden of bacterial antimicrobial resistance in 2019: A systematic analysis. The Lancet, S0140673621027240. https://doi.org/10.1016/S0140-6736(21)02724-0
Paul, M., Benuri-Silbiger, I., Soares-Weiser, K., & Leibovici, L. (2004). Beta lactam monotherapy versus beta lactam-aminoglycoside combination therapy for sepsis in immunocompetent patients: Systematic review and meta-analysis of randomised trials. BMJ (Clinical Research Ed.), 328(7441), 668. https://doi.org/10.1136/bmj.38028.520995.63
Seiler, C., & Berendonk, T. U. (2012). Heavy metal driven co-selection of antibiotic resistance in soil and water bodies impacted by agriculture and aquaculture. Frontiers in Microbiology, 3. https://doi.org/10.3389/fmicb.2012.00399
Tamma, P. D., Cosgrove, S. E., & Maragakis, L. L. (2012). Combination therapy for treatment of infections with gram-negative bacteria. Clinical Microbiology Reviews, 25(3), 450–470. https://doi.org/10.1128/CMR.05041-11
Tetard, A., Zedet, A., Girard, C., Plésiat, P., & Llanes, C. (2019). Cinnamaldehyde Induces Expression of Efflux Pumps and Multidrug Resistance in Pseudomonas aeruginosa. Antimicrobial Agents and Chemotherapy, 63(10). https://doi.org/10.1128/AAC.01081-19
Author Response
We thank the reviewer for positive comments and for the suggestions. We have revised the manuscript and included all the suggested reference - this has greatly improved the manuscript. The following list our revisions and answers to the comments and the suggestions,
- Comment: I believe that the authors should explain in an introductory section that a simple mechanism of action may just opens the way to what is possible, that positive in vitro results may transform the possible into plausible and that only in vivo data (animal model) allow to transform the plausible into probable, which ultimately will have to be confirmed by clinical trials. Indeed, a simple mechanism of action does not necessarily make a drug and even, apparently positive in vitro data, do not necessarily translate into therapeutic advances. As an example, we can cite the case of the penicillin / streptomycin combination, the synergistic character of which has been repeated for decades (including in this review lines 176-179) following the historical work of Jawetz 1953 (Gunnison et al., 1953 ) but whose clinical interest has been disputed (Paul et al., 2004) or at least debated (see the review by ( Tamma et al., 2012) ).
Answer: The following is now included in the revised manuscript and the suggested references were cited in the text,
- Line 197 – 200: We need to bear in mind that for several decades this combination principle has been tested to find a combination that is translated from in vitro to in vivo and finally into clinical combination. Not many have succeeded, though a few esp, among the β-lactam with β-lactamase inhibitor combination, and an Aminoglycoside combination.
- And in Line 191 to 195 in the revision, “On the other hand, certain combination of antibiotics has long been believed to be more effective than using a single antibiotic, however, the real effectiveness of those antibiotics combination is not clear as the resistance mechanisms continue to evolve (Gunnison et al, 1953, Tamma, et al 2012 ). Therefore, the effectiveness of those antibiotics pair or any emerging alternative strategy in multidrug environments should be continuously assessed to combat the AMR (Hegreness M, et al, 2008)”.
- Comment: We now know that the drugability of certain approaches such as bacteriophages is very difficult in particular for industrial reasons of manufacture, stability of treatments over time etc. or else, it is necessary to explain why these tracks opened 100 years ago are always promises more than realities apart from a few niche applications.
Answer: We agree with the concern of the Reviewer, and at the same time would like to retain the section on Phage therapy for the following reasons: Though phages will not be used as a first line treatment against bacterial infections as it happens with antibiotics, in a future perspective, phage therapy is expected to be applied only in clinical cases of patients who experienced the failure of antibiotic treatments. Additionally, contrary to antibiotic therapy, phage preparations for therapeutic applications are expected to be developed in a personalized way by formulating phage cocktails that might delay the emergence of bacterial resistance to phages.
- Comment: Another aspect that deserves to be highlighted is the feedback we have today on certain promises made more than 20 years ago, such as that of probiotics. At the time, the promises of probiotic promoters were only positive, but we now know that things are not so simple and that probiotics can promote the spread of AMR (this is indeed reported in this review line 526 but could be used to illustrate the fact that the safety component of new approaches must be taken into account very early in a drug development) .
Answer: We agree with the reviewer hence we have highlighted not only the positives, but also the negative outcome of the use of Probiotics, and further edited as suggested.
- Comment: In addition, Authors could point out that everything that is "natural" is not necessarily “safe” . For example, essential oils (section 4.4) can be source of resistance ( Berdejo et al., 2020) and cinnamaldehyde Induces expression of efflux pumps and multidrug resistance in Pseudomonas aeruginosa ( Tetard et al., 2019) .
Answer: Now, we have included the following text in section 4.4, line 362-366 in revised manuscript – “It is worth to highlight the demonstration of the bacteria developing resistance and tolerance towards EOs, however, cross resistance to antibiotics was not reported (Berdejo et al., 2020). Treatment of P. aeruginosa infection with sub-inhibitory concentration of cinnamon bark oil or cinnamaldehyde as adjunctive therapy may potentially induces expression of efflux pumps and this needs a further investigation to ascertain the use of EOs with any antagonistic effects (Tetard et al., 2019).
- Comment: Another aspect which is not addressed at all is environmental toxicology. We know today that the promotion of the use of copper or zinc in veterinary medicine and aquaculture was a very bad idea both for environmental toxicology (Fu et al., 2016) and for the promotion of resistance (Seiler & Berendonk , 2012) what has justified for example, the ban of the pharmacological use of Zno the EU .
We have included the following section 7.4 (Line 915-919 in the revised manuscript) – “Following the European Union - wide ban on the use of antibiotics as growth promoters, the use of copper or zinc was promoted in food animals and aquaculture. However, the use of copper or zinc results in not only damage the environment but also might promote the spread of antibiotic resistance via co-selection (Fu. Z etal., 2016, Seiler C, et.al, 2012). It is now under legislation to ban the use of ban ZnO as a veterinary medicinal product above the levels of 150 ppm in the EU from June 2022[325].
- Comment: One last major comment; I find it surprising that vaccines are viewed as " unconventional strategies ” or else the title of section 9 should begin line 870. Furthermore, it seems to me that their potential role is minimized in this review whereas according to the last seminal review just published in The Lancet , which must be cited and taken into account (Murray and al., 2022) , vaccination is a major strategy to reduce AM.R
Yes, we agree with the reviewer suggestion. The paragraph dealing on vaccines under the heading as " unconventional strategies” is moved to the section: role of vaccines in reducing the AMR pathogens. We have reorganised the sections, now in the revised manuscript, section 8 is now focussed on the role of vaccines in reducing the AMR pathogens and section 9, unconventional strategies.
As suggested, we have included Murray et al 2022. We have now included the following in the revised manuscript (line no 99 - 105), “In a first ever first comprehensive assessment of the global burden of AMR based on the statistical analysis of available data in 2019 from 204 countries, it was estimated that AMR attributes to 1.27 million deaths among the 4.95 million deaths associated with bacterial AMR. The AMR death due to resistance predicted to be highest in sub-Saharan Africa and lowest in Australasia” In 2019, MRSA were predicted to be responsible to half a million death, while the six pathogen, Escherichia coli, Staphylococcus aureus, Klebsiella pneumoniae, Streptococcus pneumoniae, Acinetobacter baumannii, and Pseudomonas aeruginosa were attributed to 50 000-100 000 deaths (Murray et al, 2022).
Minor comments
- Comment: -line 90, quote (Murray et al., 2022)
As suggested we gave quoted Murray et al., 2022 (line 105, revised manuscit).
- Comment: Several references are incorrectly numbered in the text (example ref 81 line 266 is not related to ceftiofur; line 535, reference 218 is rather 219?
Answer: Thanks for pointing the missing reference. Now, the reference 81 (line 302 in the revised manuscript) (Rajamanickam K, Yang J, Chidambaram SB, Sakharkar MK. Enhancing Drug Efficacy against Mastitis Pathogens-An In Vitro Pilot Study in Staphylococcus aureus and Staphylococcus epidermidis. Animals (Basel). 2020 Nov 15;10(11):2117. doi: 10.3390/ani10112117) is included.
In addition, we have thoroughly checked the correctness of all the references.
- Comment: -line 336: merck , not Merch ,
Answer: Line 380 in the revised manuscript- spelling of the word Merck is corrected.
This manuscript is a resubmission of an earlier submission. The following is a list of the peer review reports and author responses from that submission.